# An Infrared Array Sensor-Based Approach for Activity Detection, Combining Low-Cost Technology with Advanced Deep Learning Techniques

**DOI:** 10.3390/s22103898

**Published:** 2022-05-20

**Authors:** Arumugasamy Muthukumar Krishnan, Mondher Bouazizi, Tomoaki Ohtsuki

**Affiliations:** 1Graduate School of Science and Technology, Keio University, Yokohama 223-8522, Japan; kumar@ohtsuki.ics.keio.ac.jp; 2Faculty of Science and Technology, Keio University, Yokohama 223-8522, Japan; bouazizi@ohtsuki.ics.keio.ac.jp

**Keywords:** activity detection, deep learning, healthcare, computer vision, low-resolution, infrared array sensor, denoising, super-resolution, CGAN

## Abstract

In this paper, we propose an activity detection system using a 24 × 32 resolution infrared array sensor placed on the ceiling. We first collect the data at different resolutions (i.e., 24 × 32, 12 × 16, and 6 × 8) and apply the advanced deep learning (DL) techniques of Super-Resolution (SR) and denoising to enhance the quality of the images. We then classify the images/sequences of images depending on the activities the subject is performing using a hybrid deep learning model combining a Convolutional Neural Network (CNN) and a Long Short-Term Memory (LSTM). We use data augmentation to improve the training of the neural networks by incorporating a wider variety of samples. The process of data augmentation is performed by a Conditional Generative Adversarial Network (CGAN). By enhancing the images using SR, removing the noise, and adding more training samples via data augmentation, our target is to improve the classification accuracy of the neural network. Through experiments, we show that employing these deep learning techniques to low-resolution noisy infrared images leads to a noticeable improvement in performance. The classification accuracy improved from 78.32% to 84.43% (for images with 6 × 8 resolution), and from 90.11% to 94.54% (for images with 12 × 16 resolution) when we used the CNN and CNN + LSTM networks, respectively.

## 1. Introduction

### 1.1. Background

A new area of research is being explored: human activity recognition for intelligent healthcare. With the complications of the world’s aging population, this research field is becoming even more prominent. For instance, the elderly population in Japan in 2020 constituted 28.8% of the total population, with over 36.19 million people [1]. The aged population is the most likely to develop chronic and long-term illnesses that worsen with age [2]. Aging with chronic conditions prevents elderly people from living independently. As a result, they are reliant on social care services, such as living in nursing homes. Although some nursing systems focusing on monitoring and measuring vital signs related to the physical condition of patients have been developed, some requirements remain unsatisfied, indicating that this research field has broad application prospects.

For instance, over the last few years, the demand for non-contact monitoring of human activities has increased steadily. Non-contact activity detection systems have several advantages over ones that rely on wearable devices. In particular, their non-contact nature allows for wireless monitoring that reduces the burden on the elderly and the handicap it might cause. Non-wearable device-based systems, such as ones based on cameras, sensors [3,4], array antenna [5], Light Detection and Ranging (LiDAR) [6], Wi-Fi [7], and other similar devices, require these devices to be strategically placed in certain locations for an effective monitoring of the elderly’s activities. That being said, non-wearable devices have their own shortcomings and limitations, including privacy concerns and coverage issues.

With the development of low-cost sensing technologies, several of these issues have been addressed. For instance, the recent introduction of the wide-angle infrared (IR) array sensor helped develop device-free monitoring solutions that avoid most of the aforementioned issues. In the research related to human activity detection, two types of infrared sensors are used: pyroelectric infrared sensor (PIS) [8] and thermophile infrared array sensor [9,10]. The PIS is only capable of detecting the motion-type of human activities (i.e., activity where the person or part of their body is moving). It is not capable of detecting static human activities (e.g., when they are sitting still, standing, or laying) [11]. The thermophile IR array sensor quantifies the temperature distribution within the field of view. It is capable of detecting static and dynamic human activities, providing us with an understanding of the surrounding environment and pertinent data. Furthermore, low-resolution thermopile infrared sensors have the advantages of being small in size, low in cost, and simple to install. As a result, numerous studies have used such sensors for human activity detection, position detection, counting the number of people [12,13,14,15,16] in a room, etc.

### 1.2. Related Work

Device-free approaches for activity detection have attracted more attention over the last few years. Most of them still rely on conventional machine learning techniques. In [3,4], Mashiyama et al. proposed two similar approaches for activity detection and fall detection, respectively. In their work, they used a single IR array sensor (8 × 8 pixels) attached to the ceiling to collect the data with a fixed time window. No processing was performed on the data. However, they manually engineered four features and used them to train k-nearest neighbors (k-NN) and Support Vector Machine (SVM) classifiers, allowing them to achieve 94% accuracy. Due to the extremely high noise from the LR sensor used (8 × 8 pixels), the values for the extracted features present a high level of error. Kobayashi et al. [17] addressed some of the limitations of these works by introducing a few new features and using two sensors simultaneously, improving the overall activity detection above 90%. That being said, in their work, the authors did not address the noise issue and tested their approach in a single environment.

Javier et al. [18] proposed an approach for fall detection that relies on a single IR array sensor with a 32 × 31 resolution installed on the ceiling. In their work, the authors used conventional data augmentation techniques such as rotating and cropping the image to improve the classification accuracy up to 92%. However, due to how neural networks process the data, conventional techniques of data augmentation, such as the ones proposed here, have very little effect, thus leading to poor performance enhancement.

Matthew et al. [19] proposed an unobtrusive pose recognition using five IR array sensors with a 32 × 31 resolution. The data are collected and classified using a CNN. In their work, the authors analyzed the performance of classification of data collected by the individual sensors, as well as their combination. They achieved an overall F1 score equal to 92%. This work did not perform the classification by taking into account the temporal changes in the collected frames due to activities. Nevertheless, they used five sensors, which makes it a relatively expensive solution to justify the marginal improvements in performance.

Tianfu et al. [20] proposed a human action recognition approach using two IR array sensors. The collected data go through a set of pre-processing operations, such as quantification, time-domain filtering, and background removal. The classification was performed by a Convolutional Neural Network (CNN) and achieved an accuracy of 96.73%. However, this method fails to detect the person when (s)he is near the edges of coverage of the sensor. This is mainly due to the blurriness and noise in the images.

Miguel et al. [21] proposed a fall detection system using two IR array sensors: one having a High Resolution (HR) and the other having a Low Resolution (LR). The collected thermal data are in a fuzzy representation; the activities are classified using CNNs and the achieved accuracy is equal to 94.3%. In their work, the authors used a traditional data augmentation method to improve the classification accuracy by rotating and cropping the images. However, the authors did not combine the data, nor did they perform the sequence data classification. Furthermore, they did not consider removing the noise or enhancing the resolution of the images.

Tateno et al. [22] proposed a fall detection system using one IR array sensor placed on the ceiling. The data was pre-processed by applying noise removal and background subtraction, and activities were classified using a 3D-CNN and a 3D-Long Short-Term Memory (LSTM) separately. The highest classification accuracy reached 98.8% and 94.9%, respectively. Despite its high accuracy, this approach has not been proven to be robust as the authors did not run classification on unknown test data taken in a different environment. Furthermore, the authors used Gaussian filtering for noise removal, which is a linear smoothing filter that results in information loss.

In [23], Muthukumar et al. proposed an approach for activity detection that makes use of two IR array sensors attached to the ceiling and to the wall. The two sensors collect the data simultaneously, and their generated frames are processed together using CNN and LSTM. The CNN and LSTM were trained to perform a classification task to detect the subject’s activity. This approach achieved an accuracy of 97%. However, in their work, the authors used the raw data collected from the sensors for the classification. Such data include noisy, blurry, and distorted images, from which it is hard to identify the activity correctly in some extreme cases. While the use of the combination of data and the intensive experiments were enough to address the image distortion problem, the authors did not address other issues such as the blurriness and large amount of noise present.

A summary of the approaches described above as well as the merits and shortcomings of each is given in Table 1. As can be seen, most of the existing work did not address issues related to the noise in the data collected and works well only in particular environments. Another common limitation is the high computational and deployment cost required for some of these approaches. In addition, many of the works use two or more sensors to achieve the high performance, which, again, is expensive for real-world deployment.

### 1.3. Motivation

Most of the state-of-the-art work related to the detection of activities relies heavily on a multitude of sensors and is restricted by environmental conditions. Such activity detection systems are less effective when deployed in new, unseen environments. Nevertheless, these works, for the most part, ignore the effect of noise and image distortion on their performance. This is a key point to address as the noise level in low-resolution IR images is relatively high. It has a significant impact on the detection of activity. This presents the motivation for us to conduct this work. The current work is part of a larger project to develop a fully functional system to monitor activity [13]. In addition, it is a continuation of the research that was previously published in [12,13,23].

In this paper, we propose an activity detection system using a wide-angle infrared array sensor with advanced deep learning (DL) Computer Vision (CV) techniques. We used a single IR sensor placed on the ceiling and collected data with various resolutions (i.e., 24 × 32, 12 × 16, and 6 × 8). To faithfully increase the resolution and enhance the low quality of the collected data, we used two techniques referred to as Super-Resolution (SR) [24] and image denoising [25]. By enhancing the quality of the collected images, not only do we improve the activity detection accuracy, but we also make it more robust to changes in the environment, namely ones related to the temperature and the presence of noise sources.

We use two mainstream types of DL classifiers, namely a CNN and a combination of a CNN and an LSTM. The classification process goes as follows. First, all the individual images are classified using CNN. The CNN learns the appropriate weight in the convolutional part of the network and performs a rough classification of activities. In the second stage, the CNN’s output is passed to the LSTM, which performs a more robust classification by taking into account the temporal component. We apply quantization to the neural network to optimize the model, allowing it to run on a low-quality computational device. Nonetheless, since it is difficult to collect the data in many environments, we use a technique referred to as data augmentation [26] to generate artificial data that mimics real data. For this sake, we employ a particular type of neural network conceived for this task known as Conditional Generative Adversarial Networks (CGAN).

The use of the aforementioned DL techniques leads to a noticeable improvement in detection, as we will demonstrate throughout this paper. The contributions of this paper can be summarized as follows:We propose a lightweight DL model for activity classification that is robust to environmental changes. Being lightweight, such a model can run on devices with very low computation capabilities, making it a base for a cheap solution for activity detection.We apply SR techniques to LR data (i.e., 12 × 16 and 6 × 8) to reconstruct HR images (i.e., 24 × 32) from lower resolution ones.We use a denoising technique that requires no training to remove noise from the IR image, which significantly improves classification performance.We use an advanced data augmentation technique known as CGAN to generate synthetic data. The generated data are used as part of the training set to improve the training of the networks and generate more accurate models that are robust to environmental changes.We demonstrate that it is possible to use the LR data to achieve classification performance that is nearly identical to that of the classification of the HR data, namely 24 × 32.

The remainder of this paper is structured as follows. In Section 2, we describe in detail our proposed system and its different components, as well as our experimental setup. In Section 3, we describe the neural network architecture used for the classification, which allows for the judgment of the contributions of each of the introduced components. In Section 4, we present the results obtained by running the classification with different combinations of pre-processing. In Section 5, the main findings and possible future directions to further improve the performance of the proposed model are discussed. Finally, we conclude this work in Section 6.

## 2. Experiment Specifications

### 2.1. Device Specifications

We employ the MLX90640 IR array sensor (Designed by Melexis (https://www.melexis.com/en/product/MLX90640/, accessed on 25 April 2022) and manufactured by Sparkfun (https://www.sparkfun.com/products/14843, accessed on 25 April 2022)) depicted in Figure 1. This sensor is capable of detecting heat rays emitted by different thermal sources. The main sensor specifications are displayed in Table 2. The temperature range of the sensor covers both the typical human body temperature and the temperature of the surrounding environment. Moreover, the sensor is capable of collecting data at a variety of frame rates. In terms of the number of pixels, the sensor has a resolution of 24 × 32. The temperature rises in direct proportion to the brightness of the colors in the generated frames.

As illustrated in Figure 2, the sensor is connected to a Raspberry Pi 3 model B+. A Raspberry Pi is a single-board computer (SBC) manufactured by the Raspberry Foundation (https://www.raspberrypi.org/, accessed on 25 April 2022). The specifications of the Raspberry Pi 3 model B+ are given in Table 3. The Raspberry Pi is operated by a System on a Chip (SoC) with a quad-core ARM Cortex-A53 CPU and 1 GB of RAM. Our proposed system is meant to run on even lower-end devices. However, for the sake of this work, we run our experiments on the Raspberry Pi for its ease of operation and user-friendly operating system (OS).

In addition to the sensor, a camera is connected to the Raspberry Pi to record a video of the same event as the sensor simultaneously. The videos collected by the camera serve as the ground truth for the data collected by the sensors, allowing for easy annotation of the IR frames. While we have made sure to align the camera to the sensors on one axis and have them away from one another with a distance equal to 3 cm on the other axis using their respective boards, the alignment is not mandatory for our experiments. This is because the RGB camera is used only for the annotation of the frames, and the only constraint that needs to be satisfied is that its coverage is larger or equal to that of the IR sensor, which is indeed the case. The annotation process goes as follows: The annotator watches the frames captured by the camera and labels the frames accordingly. Since the IR sensor and the camera capture the data at the same frame rate, with negligible delay (if any), the IR sensor frames will have the exact same labels as their RGB counterparts. For example, if the annotator judges that the participants in RGB frames from 800 to 1000 are walking, this label (walking) is attributed to the IR frames from 800 to 1000 as well. We assembled the device and placed it on the ceiling. The sensor and the camera both collect the data at 8 frames per second (FPS) and store the collected frames on an SD card installed in the Raspberry Pi.

### 2.2. Environment

As described above, the system is built and attached to the ceiling at a height *h* equal to 2.6 m. The dimensions of the room and the process by which the various measurements are extracted are depicted in Figure 3.

The sensor has a wide-angle lens covering on one axis $110^{{}^{\circ}}$, and on the other, it covers $75^{{}^{\circ}}$ (we refer to the two angles as $\theta_{1}$ and $\theta_{2}$, respectively). A rectangular coverage of the following dimensions *l* and *w* was obtained at the floor level as a result.
(1)$$l=2\cdot h\cdot \tan\left(\frac{\theta_{1}}{2}\right)$$
(2)$$w=2\cdot h\cdot \tan\left(\frac{\theta_{2}}{2}\right)$$

According to the above equations, *l* and *w* are equal to 7.00 and 3.90 m, respectively. Figure 3 depicts a simplified image of a person standing at the edge of the coverage area. We can clearly see that the upper half of the body is not covered. As a result, identifying the activity (s)he would be performing when (s)he is near the edge of the measured coverage area is very difficult. Therefore, we use two coefficients $\alpha_{1}$ and $\alpha_{2}$ to obtain a more realistic coverage area, which is obviously smaller than the theoretical one measured at ground level. The empirical values of $\alpha_{1}=0.81$ and $\alpha_{2}=0.75$ are used. This practical coverage area’s length and width are $l^{\prime }$= 5.67 and $w^{\prime }$= 2.92 m, respectively.

We conducted our experiments in three separate locations.
The first room is a small, closed space with only one window that lets in little light. The temperature in the room has been set to 24 °C.The second room is larger, brighter, and equipped with an air conditioner whose temperature is set to 22 °C.In comparison to the other rooms, the third room is a little dark, and its air conditioner temperature is set to 24 °C.

Some samples of the data collected in different environments with different resolutions are shown in Figure 4.

People of various ages and both males and females, participated in the experiments. For each experiment, a single subject was asked to engage in a variety of activities for a total of five minutes in a particular environment. The sensor collects data, which we later use for classification. We managed to run several experiments and gathered enough data to train and evaluate the proposed approach.

### 2.3. Framework

A flowchart of the overall framework is shown in Figure 5. The data collected by the sensor have 24 × 32, 12 × 16, and 6 × 8 resolutions. We apply the SR and denoising techniques to the LR data. The HR data of 24 × 32 was used to generate synthetic data using CGAN to diversify the samples and cover potentially important missing samples. The synthetic data are used to train the CNN model. We classify the individual frames using a CNN. The output of the CNN was passed to the LSTM with a time window size of five frames to improve the classification accuracy. Finally, we compare all the data classification performances.

## 3. Detailed System Architecture and Description

### 3.1. Data Collection

The experiments were conducted as explained in Section 2.2, and data were collected for all the experiments run. In total, we collected 12 scenarios of data. Each scenario lasts for five minutes. The camera and the sensor both collected data at 8 frames per second.

One scenario is defined as 5 minutes of continuous activities. Each scenario includes all the activities (i.e., walking, falling, standing, sitting, lying, and the action change referred to as transition between the activities). Out of the 12 scenarios, we used 8 for training and the remaining 4 to test the model of our proposed approach. Table 4 shows the distribution of frames showing the different activities in the training and test data sets. Throughout our experiments, the scenarios chosen for the training and those chosen for testing were collected in different environments. We opted for this choice to ensure that there was no data leakage and to avoid potential problems of overfitting during the classification. In addition, for similar purposes, we made sure the activities were performed in random places rather than having them occur at the same position every time.

### 3.2. Super-Resolution

The SR technique was used on the LR data (6 × 8, 12 × 16) to learn how to upscale them back to the HR resolution of 24 × 32. By doing so, we can use low-end cheaper sensors that collect the data naively at these low resolutions (i.e., 6 × 8 and 12 × 16) and apply the trained SR model to upscale them faithfully to a higher resolution then perform the classification. By doing so, it is possible to improve the classification accuracy of frames collected by the low-end sensor to match (or get as close as possible to) the HR 24 × 32 pixel frames collected by higher-end, more expensive ones.

In our work, we use the Fast Super-Resolution Convolutional Neural Network (FSRCNN) [24] to improve image quality. The architecture of the neural network is depicted in Figure 6. It is based on a shallow network design that reproduces images faster and more clearly. The FSRCNN neural network is made up of five components:Feature extractionShrinkingNon-linear mappingExpandingDeconvolution

**Figure 6 sensors-22-03898-f006:**
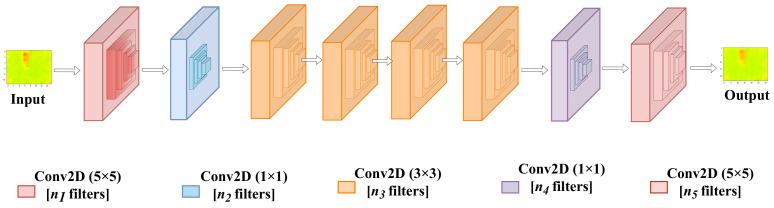
The architecture of the neural network used for Super-Resolution.

#### 3.2.1. Feature Extraction

The low-resolution image’s overlapping patches were extracted and represented as a high-dimensional feature vector. It is accomplished through the use of $n_{1}$ convolutional filters with kernel sizes equal to 5 × 5.

#### 3.2.2. Shrinking

To reduce the feature dimension, a shrinking layer was added after the feature extraction layer. This helped reduce the computational complexity. In this convolutional layer, a set of $n_{2}$ filters of size 1 × 1 was used to linearly combine the low-resolution features.

#### 3.2.3. Non-Linear Mapping

Non-linear mapping is one of the vital parts of the SR process. The purpose of non-linear mapping is to map the feature vector to a higher dimensional space. This higher dimensional space contains richer information that could be mapped to the expected output vector. In other words, it is responsible for generating enough context to reconstruct the high-resolution image. The number of sub-blocks in the non-linear mapping block and the filter size used highly influence the neural network’s performance. In our current work, we adopted a convolution layer with $n_{3}$ filters of size 3 × 3. The total number of sub-blocks is represented as m=3.

#### 3.2.4. Expanding

The shrinking layer reduces the dimension of the low-resolution feature. The quality will be poor if we generate an HR image directly from the low-resolution feature dimension. This is why we add the expanding layer after the mapping block. We used a convolution layer with $n_{4}$ filters of size 1 × 1 to maintain consistency with the shrinking layer.

#### 3.2.5. Deconvolution

A deconvolution layer was used to upscale and aggregate the previous layer’s output, resulting in a more accurate representation of the data. Unlike the convolution layer, the deconvolution layer expands the low-resolution into higher dimension data. More precisely, this is determined by the stride size, as a stride of size 1 with padding would yield information of the same size, whereas a stride of size *k* will yield condensed information of size 1/k. Deconvolution with stride expands the input data so that the output image can reach the 24 × 32 resolution.

#### 3.2.6. Activations Functions and Hyperparameters

In FSRCNN, a new activation function was introduced called Parametric Rectified Linear Unit (PReLU) for better learning. The activation threshold of PReLU is different from that of conventional ReLU. PReLU’s threshold is learned through training, whereas ReLU uses a fixed 0 as the threshold, mapping all negative values to zero. This is essential for both training and later estimating the architecture’s complexity.

We use our neural network’s total number of parameters as an indicator to estimate its complexity. To recall, our network is basically composed of a set of convolutions followed by a single deconvolution. In addition to that, we include the number of PReLU parameters.

To measure the total number of parameters of the neural network, we use the following equations that measure the total number of parameters in a convolution layer ($C_{sr}$), and the parameter in the PReLU layer ($A_{sr}$):(3)$$C_{sr}=\left(\left(m\cdot n\cdot p\right)+1\right)\cdot k,$$
(4)$$A_{sr}=h\cdot w\cdot k,$$
where *m* and *n* are the width and height of each filter, respectively, *p* is the number of channels, *k* is the number of filters used in the layer, and *h* and *w* are the input image’s height and width, respectively.

As a result, the total number of parameters in the network is 21,745 for the 8 × 6 input images and 47,089 for the 16 × 12 input images.

An example of a 24 × 32 image, its low-resolution version of 12 × 16 (resp. 6 × 8), and the reconstructed Super-Resolution one are given in Figure 7.

### 3.3. Denoising

Denoising refers to the process of restoring an image that has been contaminated by additive noise. Due to their ability to learn very fine patterns in an image, deep convolution networks have proven to be highly effective in denoising images in recent years. One of the image restoration techniques is the Deep Image Prior (DIP) [25]. This technique demonstrates that the network structure is adequate for restoring the original image from the degraded image. Pretrained networks or large image datasets are not required for this technique. It operates directly on the degraded images and learns internally what makes noise and what makes useful pixels.

Generally speaking, the most commonly used methods for image restoration in computer vision are learned prior [27] and explicit prior [28]. Learned-prior is a simple method for training a deep convolution network to learn how to denoise images by training on a data set. It takes noisy images as training data and clean images as ground truth and trains the network to reconstruct the clean image from the noisy one. In the explicit prior method, noises are mathematically calculated and removed. DIP bridges the gap between these two popular methods by constructing a new explicit prior using a convolutional neural network.

The DIP structure is based on the U-Net [29] type neural network shown in Figure 8 with multiple downstream and upstream steps and skip connections, each of which consists of a batch normalization and an activation layer. Random noise is fed into the network. The target is the image that has been tainted by the use of a mask. The loss is calculated by applying the same mask to the output image $x^{*}$ and comparing it to the noisy image. This implies that the loss function does not explicitly drive the noise/corruption repair (as it is re-applied before computing the loss). This is due to the neural network’s implicit behavior. When the network attempts to optimize toward the corrupted image. The neural network contains the parametrized weight θ. Using this θ, the network finds the optimized weight $\theta^{k+1}$ based on gradient descent optimization.
(5)$$\theta =\underset{\theta }{\mathrm{argmin}}E\left(f_{\theta }\left(z\right);x_{0}\right),$$
(6)$$\theta^{k+1}=\theta^{k}-\alpha \frac{\delta E(f_{\theta }\left(z\right);x_{0})}{\delta \theta },$$
(7)$$x^{*}=f_{\theta }\left(z\right),$$
where $x_{0}$ is the noisy image and *z* is random noise. Here *E*($f_{\theta }$ (z); $x_{0}$) is a data term usually used in the denoising problem. $f_{\theta }\left(\right)$ is a convolutional neural network-based encoder-decorder parametrized by the weight θ. The resulting denoised images are shown in the Figure 9. In our work, we applied the denoising DIP technique on 24 × 32, 12 × 16, and 6 × 8 data.

### 3.4. Conditional Generative Adversarial Network (CGAN)

When training neural networks, a common technique referred to as “data augmentation” is used to address some of the issues related to the nature and amount of data used for training. Data augmentation refers to the process of generating artificial (or synthetic) data to enlarge the size of the training set. The synthetic data improve the classification result and strengthen the system’s ability to work in various environments. The most advanced deep learning technique for data augmentation is the Conditional Generative Adversarial Neural Network (CGAN) [26]. CGAN is a generative model for supervised learning. The labeled data are used to train and generate synthetic data based on the number of classes. The CGAN structure is comprised of two neural networks: a generator *G* and a discriminator *D*, as depicted in Figure 10. *x* is the real image, and $p_{data}\left(x\right)$ and $p_{z}\left(z\right)$ denote the distribution of the real and the synthetic samples, respectively. A random noise *z* is taken from prior distributions with the label *y* and is used as an input to the generator known as a latent vector $\left(\left(z|y\right)∼p_{z}\left(z|y\right)\right)$. The generator aims to create, out of the input noise, samples with a more complex distribution G(z∣y) that are similar to that of the real ones (i.e., *x*) for the given class *y*. (x∣y) and (z∣y) are the real image with label and random noise with label, respectively. In the meantime, the discriminator should distinguish between real samples $\left(\left(x|y\right)∼p_{data}\left(x|y\right)\right)$ and the generated samples $\left(G(z|y\right)∼p_{z}\left(z|y\right))$. Backpropagation optimizations are used to train both networks, and they are completely independent of one another. The optimization of the generator is performed using the discriminator’s predictions about the samples it generated. The discriminator is trained using the generator’s synthetic data. This optimization uses CGANs’ training cost function, min–max loss, as shown in the equation below.
(8)$$\begin{array}{c} \min_{G}\max_{D}=E_{x∼p_{data}\left(x\right)}\left[\log\left(D\left(x|y\right)\right)\right]+E_{z∼p_{z}\left(z\right)}\left[\log\left(1-D\left(z|y\right)\right)\right] \end{array}$$

After several iterations of the two training techniques described above, the generator learns to generate more sophisticated samples that do resemble the real ones, and the discriminator learns how to identify the slight variation between the real and synthetic data. To reduce the cost function of each network and optimize its internal weights, a gradient step with backpropagation is performed at each iteration.

### 3.5. CNN and LSTM Classification

The classification neural network’s architecture is illustrated in Figure 11 and Figure 12. The classification consists of two stages:In the first stage, the sensor’s raw data are given as an input to the CNN that classifies the individual frames and produces the first output.In the second stage, we perform the sequence classification using the LSTM. The output of the CNN is given as an input to the LSTM with a window size equal to five frames. The LSTM produces the sequence classification output.

**Figure 11 sensors-22-03898-f011:**
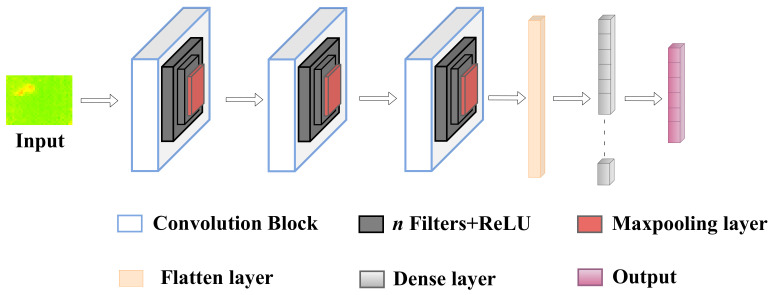
The architecture of the CNN used for classification.

**Figure 12 sensors-22-03898-f012:**
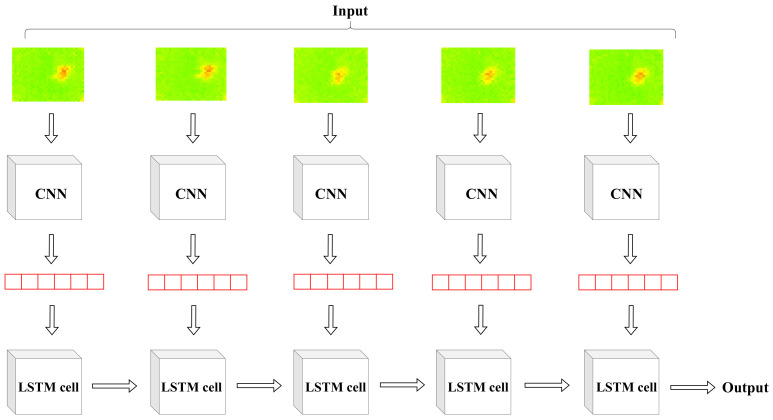
The architecture of the CNN + LSTM network used for classification.

Our neural network architecture consists of six 2D-convolution layers and two fully connected layers. Each convolution layer uses filters with a kernel size equal to 3 and has a Rectified Linear Unit (ReLu) activation function. Every two 2D-convolution layers are followed by a 2D-Maxpooling layer whose kernel size is set to 2. The output of the sixth convolution layer is flattened and is connected to a dense layer with a ReLU activation function. In the final dense layer, the activation function is sigmoid. The output of the CNN is given as an input to the LSTM network. The identical copies of CNN weights are frozen, and the trainable parameters are set to false.

To evaluate the complexity of the neural network, we measure the total number of parameters of the neural network. We use the following equations that measure the total number of parameters in a convolution layer ($C_{p}$), in a dense layer ($D_{p}$), and in an LSTM layer ($L_{p}$):(9)$$C_{p}=\left(\left(w_{f}\cdot h_{f}\cdot p\right)+1\right)\cdot c,$$
(10)$$D_{p}=\left(\left(s\cdot n\right)+1\right),$$
(11)$$L_{p}=4\times \left(\left(i_{l}+1\right)\cdot d_{l}+d_{l}^{2}\right),$$
where $w_{f}$, $h_{f}$, and *c* represent the width, height, and the number of channels of each filter, respectively, *f* represents the number of filters in the convolution layer, *s* represents the size of the dense layer, and *n* represents the number of neurons in the previous layer. The $i_{l}$ and $d_{l}$ are the input and output sizes of the LSTM neural network, respectively. The total number of parameters is about 189K in the CNN and about 568 K in the LSTM network. Compared to the existing pre-trained models, such as ResNet [30] (21M parameters) and VGG16 [31] (138M parameters), our model is lightweight and can easily run on low-end computational devices, such as the Raspberry Pi. The total number of parameters in the neural networks proposed in this work is shown in Table 5, along with that of some of the state-of-the-art neural network architectures. For its size and weight, the proposed architecture provides a very good classification performance.

### 3.6. Further Model Optimization Using Quantization

In the realm of deep learning, quantization [32] refers to the concept of using low bit-width (conventionally 8-bits) numbers to represent the weights within the neural network, rather than using floating numbers, which occupy much more space, and are more computationally costly. Operations with low bit-width numbers, such as integers, are the lightest from a computer’s perspective.

With that in mind, to achieve high accuracy for our models while keeping their computational demands as low as possible, we use this concept of quantization as introduced in [32] to reduce the size of our model. The purpose of weight quantization is to replace high weights with low weights without modifying the network’s architecture. As a result, approximated weights are used for compression. There is a trade-off between weight quantization and classification accuracy because precise weight is given up for low memory space. Weight sharing typically utilizes the same weight rather than retraining parameters. This significantly reduces computational costs. We use a quantization aware training [33,34], which has a lower loss in quantization. However, it is important to emphasize that despite its contribution to the minimization of the model size and the computation cost, the accuracy of the model when using quantization drops compared to that when using the original weights in the model after training. Quantization aware training (QAT) [34] works by applying a fake quantized 8-bit weight float to the input. The training is then operated normally as it deals with floating-point numbers, even though it emulates operations with low bit-width numbers. Once the training is complete, the information stored during the fake quantization is used to convert the floating-point model to an 8-bit quantized model.

## 4. Experimental Results

### 4.1. Computer Vision Techniques Results

To evaluate the network model’s performance in image SR and denoising, this paper utilizes a widely used image quality metric, namely the Peak Signal-to-Noise Ratio (PSNR). PSNR is commonly used to objectively evaluate image quality. It is defined as the ratio between the maximum power of the effective signal and the power of the noise in the signal. PSNR is measured in decibels (dB), and its mathematical expression is
(12)$$\mathrm{PSNR}=10\times \log_{10}\left(\frac{2^{n}-1}{\mathrm{MSE}}\right),$$
(13)$$\mathrm{MSE}=\frac{1}{mn}\sum_{i=0}^{n}\sum_{j=0}^{m}{\left|X_{ij}-Y_{ij}\right|}^{2}.$$

Here, MSE stands for the mean squared error between the original image and the generated image, which means that $X_{ij}$ and $Y_{ij}$ are the values of the pixels in the *i*-th row and the *j*-th column in the original image and the generated image, respectively. The *m* represents the numbers of rows of pixels, and *n* represents the number of columns of pixels. In general, the higher the MSE, the less similar the generated image is compared to the original; thus the PSNR decreases. In other words, a higher PSNR indicates a higher quality image.

Table 6 lists the results of Super-Resolution and denoising. As can be seen, the PSNR of the 12 × 16 frames reaches 32.62 dB. As for 6 × 8 frames, the PSNR reaches 20.47 dB. The denoising result performs well for the 24 × 32 frames; the PSNR reaches 34.62 dB. This means that the denoised frames have good quality, allowing for improving the predictions. As for the low-resolution 12 × 16 and 6 × 8 frames, the PSNR has also been improved. However, it is not enough to generate as good quality images as the HR24 × 32 ones.

### 4.2. Classification Results

We use accuracy as the metric for evaluating the efficiency of activity detection classification. Using the True Positive (TP), False Positive (FP), True Negative (TN), and False Negative (FN) values, the accuracy is calculated based on the following formula:(14)$$\mathrm{Accuracy}=\frac{\mathrm{TP}+\mathrm{TN}}{\mathrm{TP}+\mathrm{FN}+\mathrm{FP}+\mathrm{TN}}.$$

First, we report the overall classification accuracy of various techniques. Table 7 illustrates the classification accuracy achieved using the CNN. As evident, the individual applications of SR, denoising, and CGAN have helped obtain better results. The accuracy of classification using the raw data is high, it has been observed that the SR technique has further enhanced the accuracy of the result. The denoising technique has also generated better results, though the accuracy of classification of the denoised frames is less than that of SR. Furthermore, the combined application of CGAN and raw data has noticeably improved the classification accuracy from 93.12% to 95.24%. Besides experimenting with each technique aside, we applied a combination of the pre-processing techniques to the low-resolution data. As shown in Table 7, three specific combinations have been applied, namely,
Super-Resolution → Denoising,Denoising → SR,Denoising and CGAN.

**Table 7 sensors-22-03898-t007:** The overall activity classification results using CNN.

Method	Image_6×8_	Image_12×16_	Image_24×32_
Raw data	76.57%	88.22%	93.12%
SR	77.72%	89.24 %	–
Denoising	76.88%	88.30%	93.71%
CGAN+Raw data	–	–	95.24%
Denoising → SR	78.25%	89.31%	–
SR → Denoising	80.47%	91.72%	–
Denoising + CGAN	–	–	96.54%
Denoising → SR + CGAN	81.12%	92.66%	–
SR → Denoising + CGAN	83.58%	94.44%	–

Clearly, the classification accuracy of the frames enhanced using all these combinations has improved further. We further combined the pre-processing techniques with the augmented data, wherein the derived results reflect improved classification results, which are as high as that of the original HR (i.e., 24 × 32) frames. For the 6 × 8 low-resolution data, the classification results reached 83.58%, while for the 12 × 16 low-resolution data, it reached 94.44%. Remarkably, the maximum accuracy of classification of the original data with the HR (i.e., 24 × 32) has been obtained through the combined application of ‘Denoising and CGAN’, and reached 96.54%.

Further, Table 8 presents the results of sequential classification, wherein the output from the CNN is utilized as an input to the LSTM, with a time window size of five frames. Here, we observe that the sequential classification of the raw data and low-resolution data has improved considerably. By applying the Super-Resolution and denoising (both independently and collectively), the data classification results have further been improved. Further, using data augmentation techniques, we found that the sequential classification accuracy has increased.

For a fairer evaluation of our approach, we ran it against some existing approaches, namely ones that employ SVM, CNN, and 3D-CNN classifier. Table 9 compares the classification accuracy for these models with the highest accuracy of our approach. As can be seen from the table, the highest accuracy are reported in bold, our proposed approach outperforms the existing ones.

Further experiments were run to evaluate the contribution of the different image enhancement techniques. We report the activity classification accuracy using various combinations of these techniques. For simplification, and since we used different techniques, we use the following terminology for each of the resolutions or techniques used:The raw sensor data with various resolutions are referred to as $R_{\left(24\times 32\right)}$, $R_{\left(12\times 16\right)}$, and $R_{\left(6\times 8\right)}$.The SR technique applied to LR data is referred to as $SR_{\left(12\times 16\right)}$, and $SR_{\left(6\times 8\right)}$.The denoising technique applied to the HR and LR data is referred to as $DE_{\left(24\times 32\right)}$, $DE_{\left(12\times 16\right)}$, and $DE_{\left(6\times 8\right)}$.The combination of raw data with CGAN techniques is referred to as $R+CG_{\left(24\times 32\right)}$.The denoising and CGAN techniques applied to HR data are referred to as $DE+CG_{\left(24\times 32\right)}$.The combination of denoising and SR techniques applied to LR data are referred to as $DE\to SR_{\left(12\times 16\right)}$, $DE\to SR_{\left(6\times 8\right)}$, $SR\to DE_{\left(12\times 16\right)}$, and $SR\to DE_{\left(6\times 8\right)}$.The combination of SR, denoising, and CGAN techniques applied to LR data are referred to as $DE\to SR+CG_{\left(12\times 16\right)}$, $DE\to SR+CG_{\left(6\times 8\right)}$, $SR\to DE+CG_{\left(12\times 16\right)}$, and $SR\to DE+CG_{\left(6\times 8\right)}$.

The results of activity classification accuracy of CNN using the LR 6 × 8 data are shown in Table 10. We can see that each technique improved the activity classification performance. Walking and sitting, for example, present high accuracy in the $DE\to SR_{\left(6\times 8\right)}$ technique, both reaching 86%. Further, the falling accuracy is 71% in $SR\to DE_{\left(6\times 8\right)}$. Data augmentation aids in the improvement of activity detection in $SR\to DE+CG_{\left(6\times 8\right)}$ for other activities, such as standing, lying, and action change, which reach high accuracies of 84%, 81%, and 80%, respectively.

The results of the activity classification accuracy of CNN using LR 12 × 16 are shown in Table 11. Here, for the particular case of the falling activity, the $SR\to DE_{\left(12\times 16\right)}$ technique reaches a high accuracy of 94%. Similarly, the standing accuracy is 94% in the $DE\to SR+CG_{\left(12\times 16\right)}$ technique. The performance of detection of other activities improved using the $SR\to DE+CG_{\left(12\times 16\right)}$ technique, reaching a maximum of 93% accuracy in the action change activity and 92% accuracy in standing and lying.

We infer from these CNN classification results that performing SR followed by denoising and then adding CGAN data improves performance.

Table 12 shows the results of the HR data classification using CNN. Here, the $DE+CG_{\left(24\times 32\right)}$ technique performs well for the majority of the activities. Walking reaches an accuracy of 96%, sitting reaches an accuracy of 95%, and lying reaches an accuracy of 94%. The performance boost provided by denoising, which creates a clear image, aids in detecting activity.

Table 13 shows the activity classification results using CNN + LSTM on the LR 6 × 8 data. Here, again we can see that each technique improves the activity classification performance. For example, $SR\to DE_{\left(6\times 8\right)}$ has a high lying and action change accuracy of 82%. The classification accuracy of walking, standing, and sitting activities in $SR\to DE+CG_{\left(6\times 8\right)}$ is 84%, 82%, and 84%, respectively. This is thanks to the image qualify improvement after applying SR, then DE.

Table 14 shows the activity classification results using CNN + LSTM on the LR 12 × 16 data. Here, sitting and lying activities, after applying $DE\to SR+CG_{\left(12\times 16\right)}$, reached a high accuracy of 93% and 94%, respectively. Other activities, such as walking and action changes, reached 93% accuracy in $SR_{\left(12\times 16\right)}$ and $SR\to DE+CG_{\left(12\times 16\right)}$ techniques, respectively. This is thanks to the fact that the 12 × 16 resolution data contains significantly more information than the 6 × 8 resolution data. By applying the denoising technique after SR, the images are smoothed and enhanced, making it easier to detect sitting and lying activities.

The results of activity classification using CNN + LSTM applied to HR data are shown in Table 15. In most activities, the $DE+CG_{\left(24\times 32\right)}$ technique performs well. Walking has a high classification accuracy of 96%, action change has that of 97%, lying has that of 94%, and falling has that of 96%. Denoising further improves the performance by making the image clearer, which makes it easier to recognize the activity.

### 4.3. Neural Network Quantization

As previously stated, we used quantization on our neural network because one of our primary objectives is to have the proposed approach running on low-powered devices. Although the process of quantization, generally speaking, reduces the accuracy, it can still be used given flexibility for a trade-off between performance and optimization. In a first step, we compare the classification accuracy of our models with and without quantization when using the raw data (i.e., no image enhancement or data augmentation is used). Table 16 compares the performance of classification of such raw data with and without quantization. As can be seen, the accuracy is reduced, but not significantly. The accuracy drops range from 0.06% for images of resolution 12 × 16 to 2.09% for images of resolution 6 × 8. However, to recall, our main goal is to optimize the proposed deployment approach both in terms of performance and complexity. Given the different techniques proposed in this work, we compare the results of the best performing techniques with and without quantization in Table 17. As can be seen, the table shows a degradation in accuracy to some extent compared to when quantization is not used. However, this does not negate the many advantages of quantization; model sizes are reduced, and inference times are reduced to the point where they are more beneficial in low-end devices. All these results are generated using 8-bit integer data on our computer.

## 5. Discussion

In our previous work [23], we used two sensors to combine the raw data and improve activity detection. However, in the current research, we go a step further to enhance the activity detection by using a single sensor positioned on the ceiling. Utilizing a wide-angle IR array sensor with advanced deep learning computer vision techniques, we mainly aimed to develop a robust activity detection system. Describing our step-by-step approach, the previous section has presented the details of our experimental results. Herein, we observe that after applying the deep learning computer vision techniques (SR/denoising/CGAN) on the low-resolution data, we are able to generate good classification results, which are compared to the raw data classification. However, the current approach still has a few constraints to be considered and that require solutions to further enhance the activity detection ability and allow for a more general application of the proposed model. Taking into account the identified shortcomings, four key areas of improvement have been identified, as stated below:The SR technique considerably enhances the quality of the image; however, it is still challenging to detect any activity at the edge of the defined coverage area. One approach to overcome this issue is to combine SR with more advanced feature extraction methods or use a deeper neural network, for that matter.IR sensors generate the frames by collecting the IR heat rays and mapping them to a matrix. These rays are noisy by nature, which is exacerbated by the sensor’s receivers, which have a large uncertainty. This makes it hard for the IR sensors to generate high-quality data. In light of that, we used the deep learning denoising technique (DIP) to reduce the amount of noise in the image. However, the derived study results are still not up to the mark. Since this method works without any prescribed training and reduces the training cost, there is still a huge room for improvement in this direction.CGAN, one of the widely recognized advanced data augmentation techniques, was used in this research to generate synthetic data. The CGAN technique generates data based on the number of classes available and is a supervised learning technique. However, despite the use of CGAN, we found that the extraction of the features from the IR data is still very difficult, likely due to the image distortion in the raw data. It is inferred that less distortion in the raw data allows for the generation of more realistic and useful synthetic data that could contribute to much better classification performance through advanced feature extraction methods.One specific shortcoming of this research is that the only main source of heat present in the room during the experiments is the participants themselves. This means that the current model is trained for the case where there is a single person and no other heat-emitting devices, such as a stove or a computer, are present. Thus, even though the developed model performs well for the current experimental setup, it is not meant to simultaneously detect the activities of multiple persons and obstacles. A potential alternative to address this constraint could be the generation of three-dimensional data, as it could help to estimate the precise height/depth of any target individual and can also simultaneously differentiate between the activities of multiple people and objects.

## 6. Conclusions

In this paper, we have proposed an activity detection system using a low-resolution IR array sensor with advanced computer vision deep learning techniques. Using a single sensor placed on the ceiling, we have conducted a variety of experiments (primarily six human activities: walking, falling, sitting, standing, lying, and action change) in which we collected data under different conditions for a continuous period of time with different resolutions (i.e., 24 × 32, 12 × 16, and 6 × 8). To identify the activity of the participants, we ran a classification task that takes the frames generated by the sensor as the input and predicts the activity. To further enhance the classification, we applied three advanced deep learning techniques: SR, denoising, and CGAN. Herein, the key purpose was to enhance the classification accuracy of the low-resolution data. Through the results, we observed that the application of these techniques has helped improve the classification accuracy of low-resolution images from 78.32% to 84.43% (6 × 8 resolution) and from 90.11% to 94.54% (12 × 16 resolution).

## Figures and Tables

**Figure 1 sensors-22-03898-f001:**
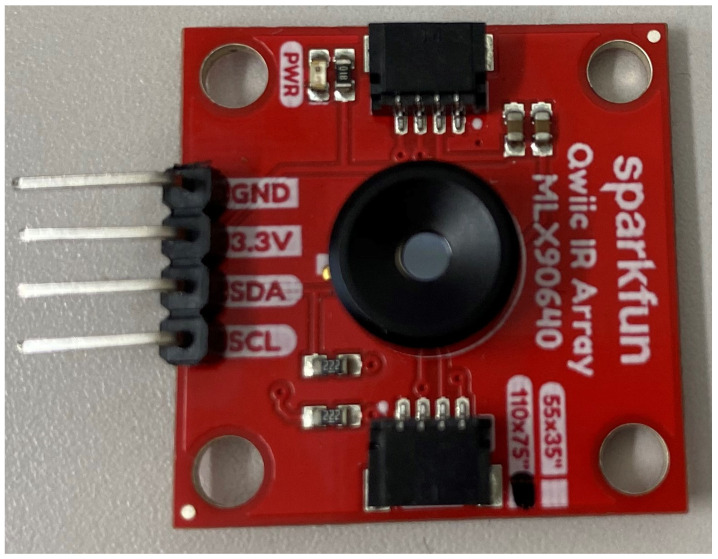
The wide-angle IR array sensor used for our experiments.

**Figure 2 sensors-22-03898-f002:**
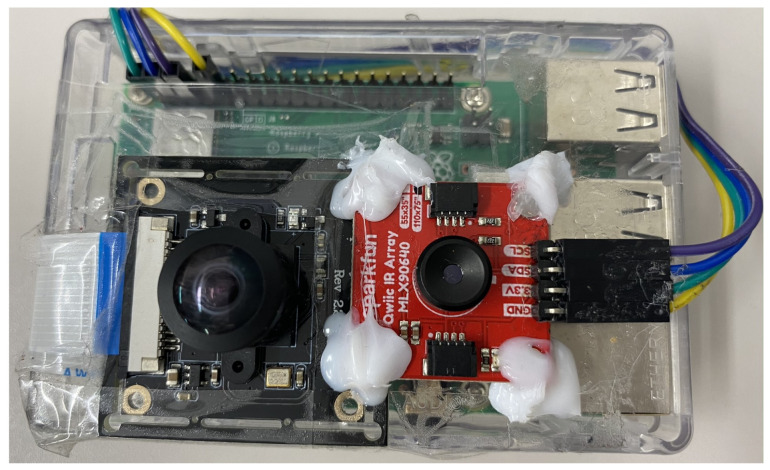
An image of the Rasberry Pi 3+ with the camera and IR sensors attached, which we used to collect data.

**Figure 3 sensors-22-03898-f003:**
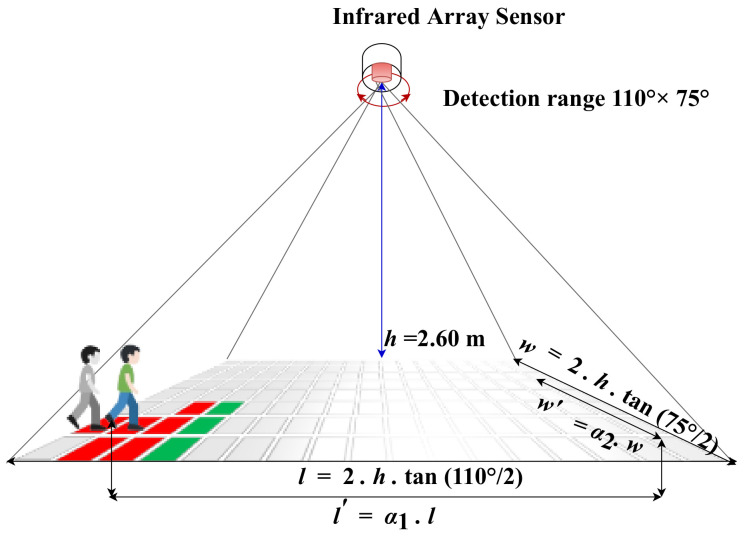
The experiment area and its dimensions.

**Figure 4 sensors-22-03898-f004:**
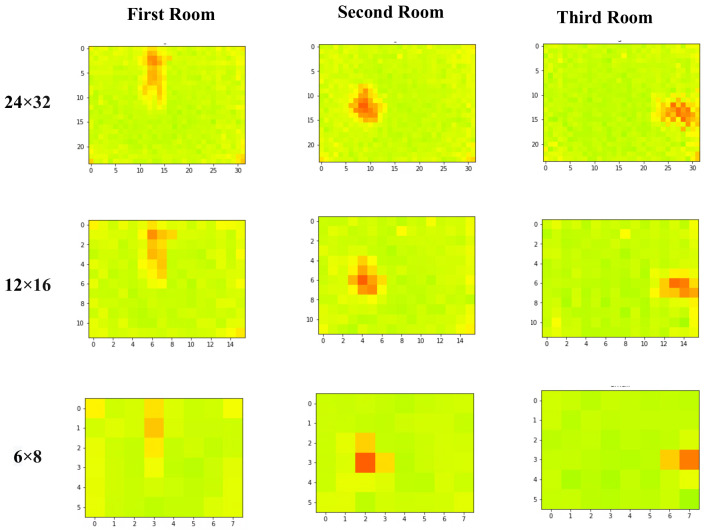
Some samples of the raw data collected in different environments with different resolutions (The red color in the image shows the human body temperature it represents the person in the room).

**Figure 5 sensors-22-03898-f005:**
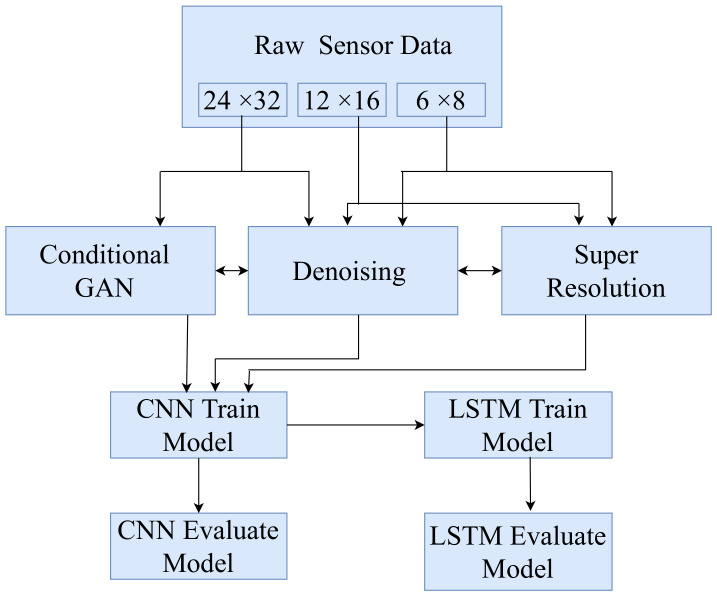
A flowchart of the proposed system.

**Figure 7 sensors-22-03898-f007:**
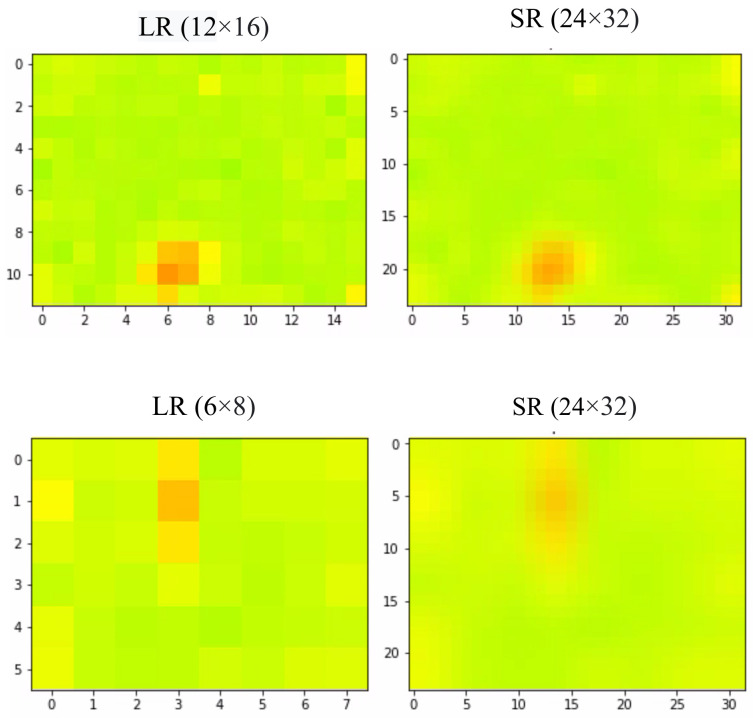
The output of the SR technique applied to a 12 × 16 frame and a 6 × 8 frame.

**Figure 8 sensors-22-03898-f008:**
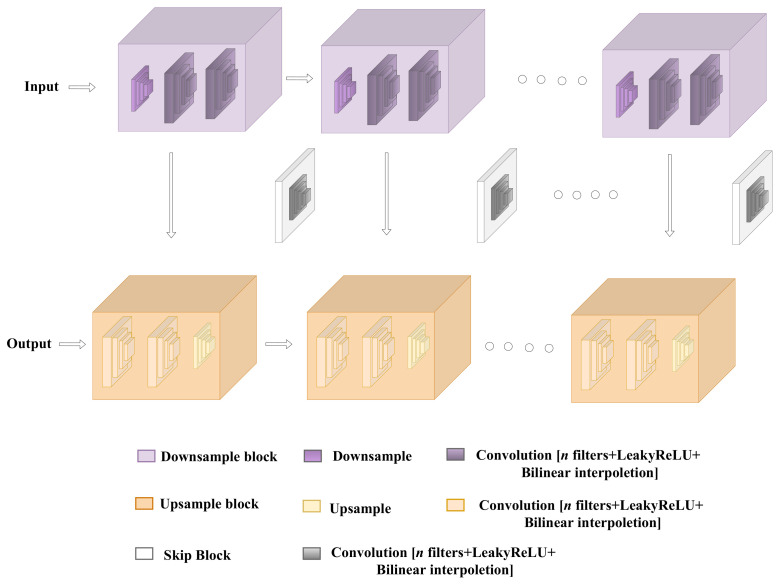
The architecture of the neural network used for denoising.

**Figure 9 sensors-22-03898-f009:**
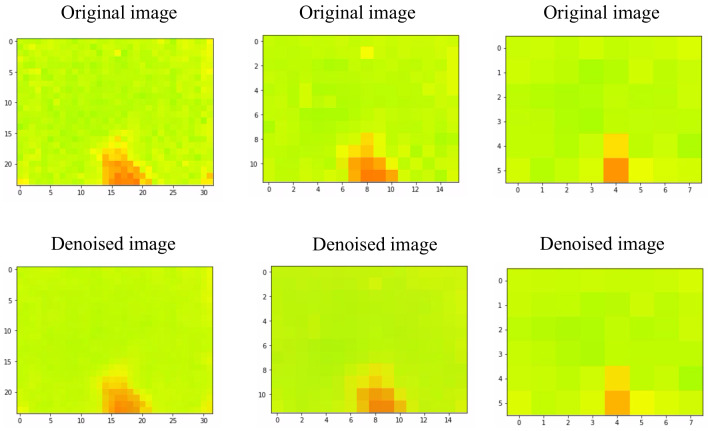
The output of denoising technique applied to 24 × 32, 12 × 16, and 6 × 8 frames.

**Figure 10 sensors-22-03898-f010:**
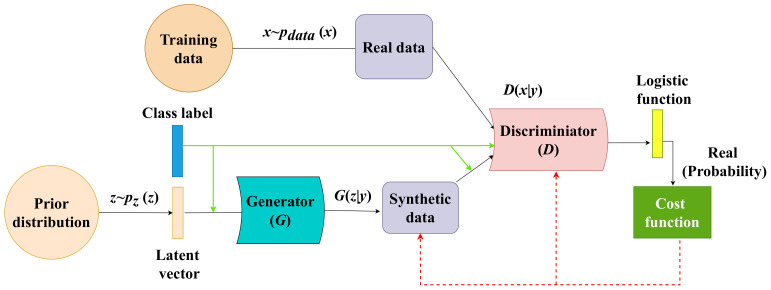
The architecture of data augmentation technique (CGAN).

**Table 1 sensors-22-03898-t001:** A summary of the existing works that use IR sensors for activity detection, their merits and shortcomings, and the obtained results.

Study	IR Sensor (Resolution)	No.of Sensors	Position of Sensor	Methods	Accuracy	Limitations
Mashiyama et al. [3]	8×8	1	Ceiling	SVM	94%	A few activities in a specific area. No pre-processing is performed. Data are highly noisy due to their low resolution.
Mashiyama et al. [4]	8×8	1	Ceiling	k-NN	94%	Due to the noise in the data, feature extraction is less effective.
Kobayashi et al. [17]	8×8	2	Ceiling, Wall	SVM	90%	No reprocessing is performed. Data are noisy. Activities are performed in very specific positions.
Javier et al. [18]	32×32	1	Ceiling	CNN	92% & 85%	Noisy and blurry image. Difficult to detect activities in high-temperature areas.
Matthew et al. [19]	32×31	5	Ceiling and all corners	CNN based on alexnet	F1-score 92%	Requires multiple sensors. Expensive to deploy in the real-world.
Tianfu et al. [20]	24×32	2	Ceiling, Wall	CNN	96.73%	Fails to detect the position of the human near the edges of coverage, due to the blurriness and noise in the images.
Miguel et al. [21]	32×31, 80×60	2	Ceiling, Wall	CNN	72%	No sequence data classification was performed. Noise removal and enhancement of the images were not performed.
Tateno et al. [22]	24×32	1	Ceiling	3D-CNN 3D-LSTM	93%	Gaussian filter is used to remove the noises, which causes a loss of information.
Muthukumar et al. [23]	24×32	2	Ceiling, Wall	CNN and LSTM	97%	Two sensors were used to detect the activity. Raw images are used for classification with lot of noise.

**Table 2 sensors-22-03898-t002:** The technical specifications of the sensor.

IR Sensor Model	Qwiic IR Array MLX90640
Camera	1
Voltage	3.3 V
Temperature range of targets	−40${}^{{}^{\circ}}$C∼85 ${}^{{}^{\circ}}$C
Number of pixels	24 × 32, 12 × 16, 6 × 8
Viewing angle	$110^{{}^{\circ}}\times 75^{{}^{\circ}}$
Frame rate	8 frames/second

**Table 3 sensors-22-03898-t003:** The technical specifications of the SBC used.

Embedded SBC	Raspberry Pi 3 Model B+
SoC	Broadcom BCM2837B0, Cortex-A53 (ARMv8) 64-bit SoC
CPU	1.4GHz 64-bit quad-core ARM Cortex-A53 CPU
RAM	1GB LPDDR2
OS	Ubuntu Mate
Power input	5 V/2.5 A (12.5 W)
Connectivity to the sensor	Inter-Integrated Circuit (I2C) serial bus
I2C transmission rate	3.4 Mbps

**Table 4 sensors-22-03898-t004:** The frame counts for each activity in the training and the testing data sets.

S.No.	Activity	Training Data Frames	Testing Data Frames
1	Walking	5456	2351
2	Standing	1959	882
3	Sitting	3102	1566
4	Lying	2486	647
5	Action change	1961	939
6	Falling	613	264

**Table 5 sensors-22-03898-t005:** A comparison between the total number of parameters of the neural networks used in the current work and those of the state-of-the-art neural networks used for image classification.

Model	Parameters
ResNET [30]	21 Million
VGG16 [31]	138 Million
CNN	189 Thousand
CNN + LSTM	568 Thousand

**Table 6 sensors-22-03898-t006:** The performance of SR and Denoising technique.

Method	Input-Output	PSNR(dB)
Super-Resolution	Image_12×16_→Image_24×32_	32.62
	Image_6×8_→Image_24×32_	20.47
Denoising	Image_24×32_	34.12
	Image_12×16_	30.52
	Image_6×8_	23.74

**Table 8 sensors-22-03898-t008:** The overall activity classification results using CNN + LSTM.

S.No.	Image_6×8_	Image_12×16_	Image_24×32_
Raw data	78.32%	90.11%	95.73%
SR	79.07%	90.89%	–
Denoising	78.55%	90.33%	96.14%
CGAN+Raw data	–	–	96.42%
Denoising → SR	80.18%	92.38%	–
SR → Denoising	82.76%	92.91%	–
Denoising + CGAN	–	–	98.12%
Denoising → SR + CGAN	80.41%	93.43%	–
SR → Denoising + CGAN	84.43%	94.52%	–

**Table 9 sensors-22-03898-t009:** A comparison between the results achieved with our proposed approach and those achieved by employing some of the existing methods in the literature.

Approach	Image_6×8_	Image_12×16_	Image_24×32_
SVM [17]	61.45%	68.52%	88.16%
CNN [21]	67.11%	82.97%	90.14%
3D-CNN [22]	72.42%	90.89%	93.28%
CNN + LSTM_SR → DE + CGAN_	**84.43%**	**94.52%**	–
CNN + LSTM_DE + CGAN_	–	–	**98.12%**

**Table 10 sensors-22-03898-t010:** The results of activity classification using CNN on 6 × 8 data.

Method	Walking	Standing	Sitting	Lying	Action Change	Falling
$R_{\left(6\times 8\right)}$	80%	68%	77%	68%	57%	55%
$SR_{\left(6\times 8\right)}$	86%	73%	86%	76%	60%	63%
$DE_{\left(6\times 8\right)}$	85%	86%	82%	68%	62%	64%
$DE\to SR_{\left(6\times 8\right)}$	86%	78%	86%	70%	70%	62%
$SR\to DE_{\left(6\times 8\right)}$	84%	82%	84%	72%	73%	71%
$DE\to SR+CG_{\left(6\times 8\right)}$	82%	80%	79%	75%	78%	70%
$SR\to DE+CG_{\left(6\times 8\right)}$	80%	84%	83%	81%	80%	68%

**Table 11 sensors-22-03898-t011:** The results of activity classification using CNN on 12 × 16 data.

Method	Walking	Standing	Sitting	Lying	Action Change	Falling
$R_{\left(12\times 16\right)}$	80%	88%	86%	75%	82%	81%
$SR_{\left(12\times 16\right)}$	84%	86%	81%	85%	84%	83%
$DE_{\left(12\times 16\right)}$	83%	85%	86%	85%	84%	79%
$DE\to SR_{\left(12\times 16\right)}$	84%	85%	89%	90%	82%	84%
$SR\to DE_{\left(12\times 16\right)}$	90%	92%	88%	91%	90%	94%
$DE\to SR+CG_{\left(12\times 16\right)}$	89%	94%	90%	85%	84%	88%
$SR\to DE+CG_{\left(12\times 16\right)}$	91%	92%	89%	92%	93%	91%

**Table 12 sensors-22-03898-t012:** The results of activity classification using CNN on 24 × 32 data.

Method	Walking	Standing	Sitting	Lying	Action Change	Falling
$R_{\left(24\times 32\right)}$	88%	90%	87%	93%	90%	91%
$DE_{\left(24\times 32\right)}$	92%	84%	92%	90%	91%	89%
$R+CG_{\left(24\times 32\right)}$	90%	95%	90%	94%	92%	92%
$DE+CG_{\left(24\times 32\right)}$	96%	92%	95%	94%	87%	93%

**Table 13 sensors-22-03898-t013:** The results of activity classification using CNN + LSTM on 6 × 8 data.

Method	Walking	Standing	Sitting	Lying	Action Change	Falling
$R_{\left(6\times 8\right)}$	75%	78%	74%	77%	70%	74%
$SR_{\left(6\times 8\right)}$	77%	74%	73%	78%	73%	76%
$DE_{\left(6\times 8\right)}$	76%	72%	78%	70%	75%	72%
$DE\to SR_{\left(6\times 8\right)}$	81%	78%	78%	75%	77%	73%
$SR\to DE_{\left(6\times 8\right)}$	78%	81%	73%	82%	82%	75%
$DE\to SR+CG_{\left(6\times 8\right)}$	80%	74%	70%	76%	78%	75%
$SR\to DE+CG_{\left(6\times 8\right)}$	84%	82%	84%	78%	81%	80%

**Table 14 sensors-22-03898-t014:** The results of activity classification using CNN + LSTM on 12 × 16 data.

Method	Walking	Standing	Sitting	Lying	Action Change	Falling
$R_{\left(12\times 16\right)}$	88%	90%	76%	77%	80%	82%
$SR_{\left(12\times 16\right)}$	85%	88%	82%	90%	86%	79%
$DE_{\left(12\times 16\right)}$	82%	86%	78%	89%	90%	83%
$DE\to SR_{\left(12\times 16\right)}$	92%	84%	77%	89%	91%	86%
$SR\to DE_{\left(12\times 16\right)}$	93%	90%	88%	84%	91%	88%
$DE\to SR+CG_{\left(12\times 16\right)}$	87%	90%	93%	94%	82%	90%
$SR\to DE+CG_{\left(12\times 16\right)}$	90%	92%	90%	86%	93%	92%

**Table 15 sensors-22-03898-t015:** The results of activity classification using CNN + LSTM on 24 × 32 data.

Method	Walking	Standing	Sitting	Lying	Action Change	Falling
$R_{\left(24\times 32\right)}$	92%	91%	93%	90%	94%	89%
$DE_{\left(24\times 32\right)}$	93%	95%	96%	91%	94%	92%
$R+CG_{\left(24\times 32\right)}$	95%	94%	95%	93%	92%	90%
$DE+CG_{\left(24\times 32\right)}$	96%	94%	93%	96%	97%	96%

**Table 16 sensors-22-03898-t016:** The performance comparison of raw data with quantization aware training.

Resolution	With Quantization	Accuracy	100 Epochs Training Time (s)	Inference Time (ms)	Model Size (MB)
6 × 8	Yes	76.23%	17	0.003	0.3
	No	78.32%	54	0.048	1.4
12 × 16	Yes	90.05%	36	0.007	0.8
	No	90.11%	88	0.078	2.4
24 × 32	Yes	94.20%	44	0.009	1.1
	No	95.73%	132	0.093	3.2

**Table 17 sensors-22-03898-t017:** A comparison between the performance of classification with and without quantization applied to the pre-processed and enhanced images using the techniques proposed above.

Resolution	With Quantization	Accuracy	100 Epochs Training Time (s)	Inference Time (ms)	Model Size (MB)
6 × 8_SR → DE + CGAN_	Yes	82.27%	145	0.38	4.18
	No	84.43%	321	2.57	10.20
12 × 16_SR → DE + CGAN_	Yes	93.18%	164	0.60	5.43
	No	94.52%	352	3.21	14.68
24 × 32_DE + CGAN_	Yes	97.53%	136	0.43	4.37
	No	98.12%	291	2.82	11.20

## Data Availability

Not applicable.

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
