# Peer review of "An Infrared Array Sensor-Based Approach for Activity Detection, Combining Low-Cost Technology with Advanced Deep Learning Techniques"

_sensors, 2022, doi:10.3390/s22103898_

Round 1
Reviewer 1 Report
This manuscript proposes an activity detection system using an infrared sensor placed on the ceiling. Using some deep learning-based methods of super-resolution and denoising enhances the quality of images. Besides, a Conditional Generative Adversarial Network is employed for data augmentation. A Convolutional Neural Network and Long Short-Term Memory combined to improve the classification performance. Extensive experiments demonstrate the effectiveness of the proposed method. However, this manuscript should add some experiments to compare with other related methods.
Here are some comments and some suggestions:
1. Please refine the problem to be solved and directly introduce the motivation.
2. Are the accuracy of the existing works compared in the same condition in table 1?
3. How to align the camera and IR sensors in your device?
4. Is the size of the data large enough to train the neural network? How to ensure that you are not over-fitting on these data?
5. Please add some experiments to compare with other related methods.
6. On page 13, line 392, "[10] For" → "[10]. For"
7. On page 5, line 209, "an an" → "an"
Reviewer 2 Report
In the authors’ previous work, they used two sensors to combine the raw data and improve the activity detection. In the current research, they went a step further to enhance the activity detection by using a single sensor, positioned at the ceiling. Utilizing a wide-angle IR array sensor with advanced deep learning computer vision techniques, they mainly aimed to develop a robust activity detection system. The key purpose was to enhance the classification accuracy of the low-resolution data. Some suggestions are as follows.
(1) The background introduction is too much. Please combine Introduction, Related Work, with Motivations and challenges into one part.
(2) What are the differences between this work and other work, such as methods, advantages, and disadvantages of performance? The authors should clearly point out the innovation of this work.
(3) Please provide the detailed type, specification, and manufacturer of involved materials and devices.
Reviewer 3 Report
This is a great work with sufficient analysis and experimental results. The weakness of this work is the number of trainable parameters in deep neural network is so high, it is suggested to add sparsity training in your work to keep the good performance and less number of parameters, such as
[1] Z. Tang et al., "Automatic Sparse Connectivity Learning for Neural Networks," in IEEE Transactions on Neural Networks and Learning Systems, doi: 10.1109/TNNLS.2022.3141665.
[2] K. Zhang, et al., "Deep Sparse Learning for Automatic Modulation Classification Using Recurrent Neural Networks", sensors, 2021.
[3] Cuculo, V.; D’Amelio, A.; Grossi, G.; Lanzarotti, R.; Lin, J. Robust Single-Sample Face Recognition by Sparsity-Driven Sub-Dictionary Learning Using Deep Features. Sensors 2019, 19, 146. https://doi.org/10.3390/s19010146
2. If you don't use sparse neural network, you may consider to quantize the weights of your neural network to reduce significantly the size of your design. For example,
[1] Huang, Q. Weight-Quantized SqueezeNet for Resource-Constrained Robot Vacuums for Indoor Obstacle Classification. AI 2022, 3, 180-193. https://doi.org/10.3390/ai3010011
[2] Huang, Q.; Hsieh, C.; Hsieh, J.; Liu, C. Memory-Efficient AI Algorithm for Infant Sleeping Death Syndrome Detection in Smart Buildings. AI 2021, 2, 705-719. https://doi.org/10.3390/ai2040042
Reviewer 4 Report
This paper presents an activity detection system using infrared array sensor, placed on the ceiling. The authors achieved the results and presented well. I suggest authors to add some more paragraphs by giving citations in the introduction section.
Round 2
Reviewer 2 Report
The authors have responded to most of my concerns. There is no other suggestion.
Reviewer 3 Report
The quality of this work has been improved